# *Myzus persicae* Management through Combined Use of Beneficial Insects and Thiacloprid in Pepper Seedlings

**DOI:** 10.3390/insects12090791

**Published:** 2021-09-03

**Authors:** Qingcai Lin, Hao Chen, Xiaoyan Dai, Shuyan Yin, Chenghao Shi, Zhenjuan Yin, Jinping Zhang, Feng Zhang, Li Zheng, Yifan Zhai

**Affiliations:** 1Institute of Plant Protection, Shandong Academy of Agricultural Sciences, Jinan 250100, China; linqingcai@yeah.net (Q.L.); cha.active@gmail.com (H.C.); 15169087554@163.com (X.D.); sch0626@163.com (C.S.); zhengli64@126.com (L.Z.); 2College of Plant Protection, Shandong Agricultural University, Tai’an 271000, China; shuyany@163.com; 3College of Agriculture, Guizhou University, Guiyang 550025, China; yinzhenjuan1220@126.com; 4MoA-CABI Joint Laboratory for Bio-Safety, Institute of Plant Protection, Chinese Academy of Agricultural Sciences, Beijing 100193, China; J.Zhang@cabi.org (J.Z.); F.Zhang@cabi.org (F.Z.)

**Keywords:** *Myzus persicae*, integrated pest management, neonicotinoid insecticides, *Harmonia axyridis*, *Aphidoletes aphidimyza*, *Bombus terrestris*

## Abstract

**Simple Summary:**

*Myzus persicae* is a worldwide pest causing significant economic loss, especially to vegetables. However, the mainly applied insecticides were not effective, whilst also endangering the safety of pollinators. *Harmonia axyridis* and *Aphidoletes aphidimyza* are predators of aphids, but they are costly and affected by temperature and insecticides. We conducted toxicity tests and greenhouse trails to make an effective combination of neonicotinoid insecticides and predators. Both *H. axyridis* and *A. aphidimyza* effectively controlled aphids whether combined with thiacloprid or not, at above 20 °C temperature condition. Our results indicated that it is it is necessary to choose *H. axyridis* or *A. aphidimyza* to control aphids based on economic and thermal considerations. Practically, thiacloprid could be used either as an emergency option to control aphids’ abundance alone or in combination with natural enemies.

**Abstract:**

*Excessive* insecticide application has posed a threat to pollinators and has also increased insecticide resistance of *Myzus persicae* Sulzer. Therefore, it is urgent to develop an economical and effective strategy, especially for greenhouse vegetables. Firstly, we selected a neonicotinoid insecticide that is specifically fatal to *M. persicae* but relatively safe to predators and bumblebees by laboratory toxicity tests and risk assessments. Then, we tested the effectiveness of the neonicotinoid insecticide under different temperature conditions. According to the LC_50_ values and the hazard quotients, thiacloprid met the requirements. Greenhouse trails indicated that thiacloprid was quite efficient, while control dropped to 80% without the application of thiacloprid. As for biological control, *Harmonia axyridis* effectively controlled 90% of aphids with thiacloprid or not. However, *Aphidoletes aphidimyza* performed better above 20 °C. Our results indicated that it is cost-effective to control *M. persicae* with *A. aphidimyza* in suitable temperature conditions and *H. axyridis* was more effective at low temperatures. Practically, thiacloprid could be used either as an emergency option to control aphids’ abundance alone or in combination with natural enemies.

## 1. Introduction

The green peach aphid, *Myzus persicae* (Sulzer) (Hemiptera: Aphididae), is a worldwide economically important pest with a host range of over 400 plant species [1]. *M. persicae* inflicts serious damage on plants, including vegetables grown in the greenhouse, directly by ingesting phloem or indirectly transmitting over 100 different plant viruses [1]. Populations of *M. persicae* can increase rapidly and cause serious damage in a short period of time due to its continual parthenogenesis and short generation time. The control of *M. persicae* has exclusively relied on chemical insecticides, making it become one of the most widely resistant pest [2]. Integration of biological and chemical control methods have been conducted by use of selective active ingredients that are safe for the beneficial insects, with reduced dose rate, then predators would prey upon the aphids that have survived the insecticide [3]. Therefore, more effective chemical and natural enemy combinations for *M. persicae* control need to be practiced.

Neonicotinoid insecticides (neonics) are the most effective among insecticides to control pests that damage plants by sucking, such as hemipterans and thysanopterans. During the 1990s, imidacloprid became the first commercial neonicotinoids. Since then, other kinds of neonicotinoids were gradually synthesized, including thiamethoxam, acetamiprid, thiacloprid, clothianidin, dinotefuran, nitenpyram, sulfoxaflor, flurofuranone and triflumezopyrim [4]. Neonicotinoids had been widely used in the world for nearly 30 years, which has led to the development of resistance, for example, the *M. persicae* populations can exhibit molecular and behavioral resistance soon after the intensive introduction of neonicotinoids [2,5]. Moreover, some studies revealed that extensive use of neonicotinoids have caused adverse actions to pollinators, ants and insectivorous birds [6,7,8]. In outdoor crops’ pest control, neonicotinoids use is now severely restricted in Europe [9]. Nevertheless, neonicotinoids are still globally important insecticides to control some sucking-type pests in the greenhouse. In order to gain the better advantages of neonicotinoids, and avoid the adverse actions to beneficial insect problems, neonicotinoids integration with natural enemies may be a much safer and efficient pest control strategy in greenhouses.

*Harmonia axyridis* (Pallas) (Coleoptera: Coccinellidae), native to Asia, is a predator of small arthropods, especially aphids, of numerous species in natural and managed landscapes. Both larvae and adults tend to aggregate and prey on aphids, while the aphid consumption increased with larval instar, increasing prey aggregation and density [10]. *H. axyridis* has been widely applied to control *M. persicae* and other aphid species in a wide range of ecosystems, such as trees, field crops (i.e., wheat and cotton fields) and greenhouse vegetables [11]. However, because of its aggregation and remarkable expansion, some countries considered it as an invasive species, while focusing its unfavorable (negative) effects on the environment, in general, and particularly on ladybird diversity [11,12]. *Aphidoletes aphidimyza* (Rondani) (Diptera: Cecidomyiidae) is a generalist aphid predator that can feed on 85 aphid species [13]. Females can locate aphid colonies and lay eggs in a 45 m radius efficiently [14]. Only larvae can prey on aphids, leaving shriveled aphid bodies attached to plants, especially the older larvae that have higher predation rate. Moreover, the number of aphids killed varies with prey density (i.e., more aphids were killed than predators nutritionally needed when in high aphid densities) [15,16]. *A. aphidimyza* have been commercially released in greenhouse crops such as sweet pepper, cucumber, eggplant, potted ornamentals and woody ornamentals in North America and Europe [17]. Another ecological agricultural technical measure—bumblebee pollination can increase the quality and yield of pollinated crops for greenhouse crops. *Bombus terrestris* (Linnaeus) (Hymenoptera: Apidae, Bombini) is one of the important commercially available pollinators, and it is very sensitive to pesticides, especially some high risk pesticides [6,18,19].

The goal of this study was to select a suitable neonicotinoids to integrate with natural enemies to develop a lasting and cost-effective strategy to control the notorious pest *M. persicae* without endangering pollinators in the greenhouse. Therefore, we assessed the risk of neonicotinoids with *B. terrestris*, *H. axyridis* and *A. aphidimyza*, and validated thiacloprid suitability either with *H. axyridis* or *A. aphidimyza* for the control of *M. persicae* through a series of laboratory assays and a greenhouse efficacy trial.

## 2. Materials and Methods

### 2.1. Insects and Insecticides

Initial population of *M. persicae* colony was established by collecting its population in 2019 with different life ages from pepper plants grown in field near Ji’nan in Shandong Province, China. They were reared on tobacco seedlings in the laboratory under a temperature of 25 ± 2 °C, photoperiod of 16L:8D (h) and 70 ± 5% relative humidity. *A. aphidimyza* pupae and *B. terrestris* were obtained from Shandong Lubao Technology Co. Ltd. (a specialized manufacturer of beneficial insects in Jinan, China). Eggs of *H. axyridis* were bought from Henan Jiyuan Baiyun Industrial Co. Ltd., Jiyuan, China. Both *A. aphidimyza* and *H. axyridis* were reared to second instar with pea aphids (*Acyrthosiphon pisum* Harris) (Hemiptera: Aphididae) while *B. terrestris* workers reared on the same fresh pollen diet and sugar syrup were used for acute toxicity experiment.

The neonicotinoid insecticides include imidacloprid WG (70%, Bayer Crop Science, Leverkusen, Germany), nitenpyram AS (10%, Zhejiang Shijia Technology Co. Ltd., Deqing, China), acetamiprid SP (20%, Jiangsu Longdeng Chemical Co. Ltd., Suzhou, China), thiacloprid SC (2%, Shandong Guorun Biological Pesticide Co. Ltd., Taian, China), thiamethoxam WG (25%, Syngenta Crop Protection, Basel, Switzerland), clothianidin SC (20%, FMC Corporation, Philadelphia, PA, USA), dinotefuran SG (20%, Mitsui Chemicals AGRO, Tokyo, Japan) and flupyradifurone SL (17%, Bayer Crop Science, Leverkusen, Germany) were used. All pesticide stock solutions were prepared in water (without a carrier solvent) immediately prior to use.

### 2.2. Acute Toxicity Determination

For aphids, the aphid-leaf-dip bioassay was conducted according to Srigiriraju et al. [20] with modifications. Briefly, fresh tobacco leaves with at least 20 aphids (4–5 instar) were dipped for 10 s in the designated concentrations (Appendix A), air dried and placed on slightly moistened filter papers in plastic cups. Aphid mortality was assessed at 48 h after exposure and aphids were considered dead when they did not move after lightly touched with a fine paintbrush.

For *A. aphidimyza* and *M. persicae*, residual film in glass tubes were conducted according to Lin et al. [21]. The 60 mL circular glass tubes (3 cm diameter, Jinan Huihengtong Co. Ltd., Jinan, China) filled with 1 mL designated concentrations were kept stirred for approximately 2 h to generate residual film. The 20 s instar larvae of *A. aphidimyza* with *M. persicae* were introduced to each glass tube. Mortality of the larvae were recorded after 48 h when they remained immobile after being touched with a fine paintbrush. The method to test toxicity of pesticide formulations on *H. axyridis* was similar to that used for *A. aphidimyza*, except that only one larva was introduced in each glass tube, and there were 10 tubes for each concentration.

The bumblebee test scheme was modified based on previous research about honeybees [22]. For this, newly emerged workers were collected from the *B. terrestris* colony, the different concentrations of those selected neonicotinoids with Tween-80 were dropped onto the mesonotum of workers using the microapplicator (Burkard, Rickmansworth, UK). After the liquid become dry, ten workers were placed in an artificial stainless steel nest box (14 cm × 7 cm × 10 cm, Jinan Huihengtong Co. Ltd., Jinan, China), and reared on the sugar syrup for 48 h to obtain the death rate data.

Acute LC_50_ bioassays were performed with 5 or 6 doses (Appendix A) or 8 neonicotinoid insecticides and 3 replication. All the *A. aphidimyza* and *H. axyridis* treatments were kept at 25 ± 2 °C, 70 ± 5% relative humidity (RH) and a photoperiod of 16L:8D (h). However, *B. terrestris* treatments were kept under conditions of 25 ± 2 °C, 60 ± 5% RH and continuous darkness. Obtained data were corrected for control mortality (which was never larger than 10%) using Abbotts’s formula before analyses. The slope, LC_50_/LD_50_, 95% confidence interval and LR_50_ were estimated, and correlation coefficients was performed.

### 2.3. Risk Assessment Procedures

Assessment procedure for *A. aphidimyza* and *H. axyridis* was based on the environmental risk assessment guidelines for non-target arthropods [23,24], which has been described by Lin et al. [22]. The in-field predicted exposure rate (PER _in-field_) = the application rate × the multiple application factor (MAF), the hazard quotient (HQ _in-field_) = PER _in-field_/the application rate for 50% mortality (LR_50_). HQ _(in-field)_ < 2 indicates low risk, high risk which need higher tier tests if not [25,26].

The assessment of *B. terrestris* followed the HQ mode of EPPO [27]. It was considered to be low risk if HQ ≤ 50, medium risk if 50 < HQ ≤ 2500 and high risk if HQ ≥ 2500. Semi-field tests would be triggered when risk was medium or high.
(1)$$\mathrm{HQ}=\mathrm{application}\;\mathrm{rate}\;\left(g\;a.i.\;ha^{-1}\right)/{\mathrm{LD}}_{50}\;\left(\mu g/\mathrm{bee}\right)$$

### 2.4. Greenhouse Efficacy Trial

Greenhouse efficacy trials were conducted on pepper seedlings using cages (90 cm × 90 cm × 90 cm, Zhangjiagang Phoenix red light science and education equipment factory, Zhangjiagang, China) in a greenhouse from 29 October to 25 November 2019. At first, on 29 October, thirty adults of *M. persicae* were released on the leaves of each pepper seedling (15–20 cm height) by a soft brush. Then, 30 pepper seedlings were set in one cage (90 cm × 90 cm × 90 cm), each treatment had two cages as replicates. Aphid population was recorded at five-day intervals during the study period. The following six treatments were established: (I) Thiacloprid: Pepper seedlings treated with the recommended concentration of acetamiprid (20 mg a.i.·L^−1^); (II) *H. axyridis*: One newly hatched larva was introduced to each seedling by a soft brush; (III) low-dose thiacloprid and *H. axyridis*: Pepper seedlings treated with the concentration of LC_50_ to *M. persicae* (0.04 mg a.i.·L^−1^); 1 newly hatched larva of *H. axyridis* was introduced per 2 seedlings; (IV) *A. aphidimyza*: Five newly emerged female adults of *A. aphidimyza* were introduced to each cage; (V) low-dose thiacloprid and *A. aphidimyza*: Pepper seedlings treated with the concentration of LC_50_ to *M. persicae* (0.04 mg a.i.·L^−1^); three newly emerged female adults of *A. aphidimyza* were introduced to each seedling; (VI) control group: Water spray only. Thiacloprid or water was sprayed with Amway spray bottles (Amway China, Guangzhou, China) until droplets were evenly distributed on the leaves, ensuring that the inner and outer stems and all leaves were evenly treated. Thiacloprid or water was sprayed at 4:00 p.m. on 31 October and 7 November. Adults of *A. aphidimyza* were introduced at 4:00 p.m. on 29 October and 5 November, while larvae of *H. axyridis* were introduced at 4:00 p.m. on 1 November and 5 November.

Due to the low temperature, the aphid stocks in *A. aphidimyza* and low-dose thiacloprid combined *A. aphidimyza* treatments were significantly higher; therefore, we conducted these two treatments at a higher temperature from 28 April to 30 May 2020. On 28 April, thirty adults of *M. persicae* were introduced. Thiacloprid or water was sprayed on 30 April and 7 May and newly hatched larva of *H. axyridis* or adults of *A. aphidimyza* were introduced on 1 May and 8 May, while the population of aphids was recorded every fifth day from 30 April. The temperature fluctuation recorded during the tests is shown in Appendix A, the mean temperature was 19.9 °C from 29 October to 25 November 2019, and was 24 °C from 28 April to 25 May 2020.

The *M. persicae* population reduction rate and percentage reductions compared to a control were computed as following [28]:(2)$$\mathrm{Population}\;\mathrm{reduction}\;\mathrm{rate}\;\left(\%\right)=\frac{\left(e-a\right)}{e}\times 100\%$$
(3)$$\mathrm{Control}\;\mathrm{effect}\;\left(\%\right)=\frac{\left(PT-PC\right)}{\left(100-PC\right)}\times 100\%$$
where *e* = pest number at day 1, *a* = pest number after day 1, *PT* = population reduction rate of treated group, *PC* = population reduction rate of control group. The total number of larvae, pupae and adults of *H. axyridis* and *A. aphidimyza* in each treatment were counted.

For the interaction between treatments and temperatures, the ‘fitdistr’ and ‘AIC’ functions in R were used to identify error distributions. Then, two-way ANOVA was used to analyze the population numbers of *M. persicae*. Tukey’s honestly significant differences (HSD) was used as post hoc test (*p* < 0.05) when significant differences were detected.

## 3. Results

### 3.1. Acute Toxicity of Neonicotinoids to Insects

The statistical results of the acute toxicity regression equation including the LC_50_ together with their 95% confidence intervals with good fit of the data and linear regression models (r^2^ > 0.9) are shown in Table 1. According to LC_50_, we found that *A. aphidimyza* was the most sensitive to the eight neonicotinoids with LC_50_ ≤ 0.34 mg a.i.·L^−1^. *H. axyridis* was also significantly more sensitive to most of the tested neonicotinoids than *M. persicae* except to nitenpyram and thiacloprid, whose LC_50_ values were 17.067 and 1.314 mg a.i.·L^−1^ higher than *M. persicae*, respectively. *B. terrestris* was more sensitive to nitenpyram than thiacloprid at 24 or 48 h, the LD_50_ values were <0.6 and >17 µg a.i.·bee^−1^, respectively (Table 2).

### 3.2. Risk Assessment of Pesticides to Beneficial Insects in Field

The data of maximum field recommended rates, number of applications and interval of applications in the field were obtained from the China pesticide information network (http://www.icama.org.cn (accessed on 29 July 2019)). As illustrated in Table 3, all of the HQs _(in-field)_ of *A. aphidimyza* were much bigger than 2, indicating the risks of eight neonicotinoids to *A. aphidimyza* were high risk. Low risks of nitenpyram and thiacloprid to *H. axyridis* (HQ < 2) were consistent with acute toxicity results. Two-time points of HQ of nitenpyram to *B. terrestris* were more than 50; however, the HQ values of thiacloprid were much less than 50 (Table 4). For the purpose of the following experiment, thiacloprid was selected due to its absolutely low risk to *H. axyridis*, *B. terrestris* and relatively low risk to *A. aphidimyza*.

### 3.3. Greenhouse Efficacy Trial

The population size of *M. persicae* during the trial showed increased aphid population in control groups (Figure 1). The number of *M. persicae* at the sixth investigation in high temperature (25 May, averaged 24 °C) was 2.3 times of that in low temperature (25 November, averaged 19.9 °C) (*p* < 0.001). No matter whether under low temperature or higher temperature conditions, the chemical control effect is remarkable at the beginning, the highest control effects were 97.38% and 99.39%, respectively (Table 5). However, as the use of pesticides stopped, the number of insects began to rise, leading to continuous reduction in the control effects. Since the third investigation, *H. axyridis* controlled the number of *M. persicae* to 1.13 and 2.68 in low (10 November) and high temperature (10 May) conditions, respectively, and kept the control effects over 96%. There were no significant differences between the two temperature conditions in insecticide and *H. axyridis* treatments (*p* = 0.14, *p* = 1.00). *M. persicae* population can also be controlled effectively when reduced numbers of *H. axyridis* were applied together with a low dose of thiacloprid; the control effects increased to over 90% in both low and high temperature conditions (Table 5). When in low temperature, the control effects in groups of *A. aphidimyza* and *A. aphidimyza* + thiacloprid were 79.41% and 50.87%, respectively. However, the *M. persicae* population in groups of *A. aphidimyza* and *A. aphidimyza* + thiacloprid continuously declined when the temperature was higher, with control effects of 90.00% and 97.79% on 25 May, respectively. The interactive effects of treatments and temperatures were found to be significant according to two-way ANOVA (Table 6).

Highest numbers of *H. axyridis* or *A. aphidimyza* in each treatment were recorded in high temperature conditions (Figure 2 and Figure 3). This situation was more remarkable in *A. aphidimyza* application cages; there were no more than four *A. aphidimyza* in low temperature condition, whereas 367 larvae and adults were in high temperature condition. It is worth noting that the numbers of *H. axyridis* in each condition were much less than originally released (e.g., in *H. axyridis* treatment, there were only four out of 60 *H. axyridis* on 25 May).

## 4. Discussion

Although, overreliance on neonicotinoids would lead to the development of resistance in the *M. persicae* population [29], the neonicotinoids are effective to control *M. persicae* in this study (Table 1 and Table 5, Figure 1). Therefore, the dosage and frequency of neonicotinoids should be strictly followed to avoid or slow down resistance of *M. persicae.* Furthermore, chemical insecticides cannot provide lasting control of *M. persicae* (Figure 1). Therefore, it is necessary to integrate neonicotinoids with natural enemies to control *M. persicae* effectively and sustainably.

Natural enemies of aphids are not necessarily more susceptible to insecticides than their aphid prey [3]. Here we found that nitenpyram and thiacloprid showed more toxicity to *M. persicae* as compared to *H. axyridis* (Table 1 and Table 3). However, larvae of *A. aphidimyza* were more sensitive to neonicotinoids (Table 1), and all of the neonicotinoids assessed were higher risk to *A. aphidimyza,* while the HQ _(in-field)_ of thiacloprid was among the lowest (Table 3). Boulanger et al. [13] summarized that neonicotinoids are generally toxic to *A. aphidimyza,* while larvae are more sensitive than adults to spray applications of pesticides. It would be relatively safe since the low-dose thiacloprid (0.04 mg a.i.·L^−1^) used in the field trial was lower than the LC_50_ (0.13 mg a.i.·L^−1^) of *A. aphidimyza*. Furthermore, dose reduction encourages the appearance of insecticide tolerance genotypes; however, predators would prey on the aphids that have survived the insecticide [3]. In this study, *M. persicae* were consumed by *H. axyridis* at both temperatures, and by *A. aphidimyza* at higher temperature.

Many researchers show that wide use of some neonicotinoids will seriously endanger pollinator colonies, such as the honeybee and bumblebee [9,30]. In the past few years, some governments have issued their relevant policies to restrict or inhibit some neonicotinoids use outside of permanent greenhouse structures [31]. At present, the bumblebee pollination technique has increasingly been prevalent all over the world to increase the quality and yield of greenhouse crops. According to the acute toxicity and risk assessment, thiacloprid was low risk to *B. terrestris* (Table 2 and Table 4), thus thiacloprid would be safe to *B. terrestris* that were globally applied for vegetables pollination in greenhouses.

The control effects of *H. axyridis* and *H. axyridis* + thiacloprid were efficient both at lower and higher temperatures (Table 5, Figure 1). *H. axyridis* is an excellent prospective biological control agent of aphids, especially the fourth-instar larvae and adults, which makes the mass production of commercially available *H. axyridis* costly [32,33,34]. Furthermore, due to its negative effects on the ladybird diversity in Europe and America, *H. axyridis* should be released cautiously [12,35]. *A. aphidimyza* are distributed widely throughout the world except Australia and Polynesia, and one larva preying on less than 10 aphids can complete its development [13]. Under high temperature conditions, the highest number (n = 367) of *A. aphidimyza* was much higher than *H. axyridis* (Figure 3), indicating that it can establish a stable population more easily and control *M. persicae* effectively and sustainably. Thus, *A. aphidimyza* are a more cost-effective predator and ecological secure when commercially used.

The control effects of *A. aphidimyza* and *A. aphidimyza* + thiacloprid at higher temperature were efficient; however, could not control the population of *M. persicae* when the temperature was lower than 20 °C. Applications of *A. aphidimyza* resulted in an over 90% reduction rate of *M. persicae* at average daily temperatures between 20 to 25 °C [17]. But predation, developmental period, survival and fecundity rates of *A. aphidimyza* decreased at lower temperatures (<20 °C) [36]. Additionally, the egg hatching, larval and adult longevity were negatively influenced when constantly reared at 35 °C [37]. Therefore, the temperature and timing of release are important factors that influence aphid control by *A. aphidimyza*.

The efficacy of aphid control by *A. aphidimyza* would be improved when combined with parasitoids *Aphidius colemani* [38]. Specialist natural parasitoids are often found in fields and greenhouses. When *A. aphidimyza* were applied alone or integration with low-dose thiacloprid, the natural parasitoids would be protected and contribute to aphid control. *A. aphidimyza* control effects would be weakened, since they are always the one who will be preyed on by other predators, especially by *Orius* and ladybirds [39,40]. Therefore, predators must be selected carefully when two or more pests exist in one greenhouse.

To our knowledge, this is the first report on the integrated control of *M. persicae,* combining the predatory insects *H. axyridis* and *A. aphidimyza* with the selected low-risk neonicotinoids. Our results indicate that IPM methods using a low density of predatory insects with low doses of neonicotinoids could control the population density of *M. persicae* for longer periods, which proved low risk for *B. terrestris*. Overall, the use of thiacloprid in combination with *H. axyridis* or *A. aphidimyza* can provide an effective control of *M. persicae*. *H. axyridis* provided a better control effect as compared to *A. aphidimyza,* especially in low temperature conditions. The cost of the IPM strategy based on the chemical and biological control methods was similar to the cost of chemical control. Whether due to the control effect or input cost, cooperative control of *M. persicae* in greenhouse vegetables using the IPM methods has practical application prospects for farmers.

## Figures and Tables

**Figure 1 insects-12-00791-f001:**
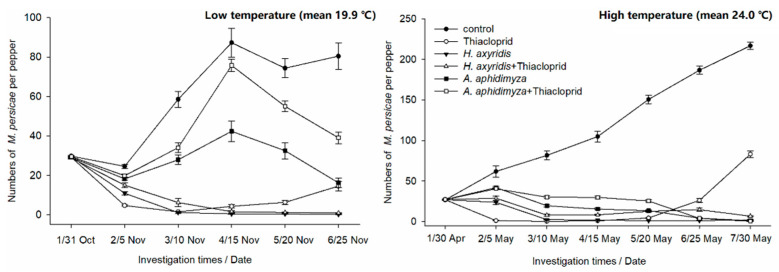
Population size of *M. persicae* on pepper seedlings with different predators and insecticide treatments within a greenhouse. Mean number of *M. persicae* (±SE) in low and high temperature.

**Figure 2 insects-12-00791-f002:**
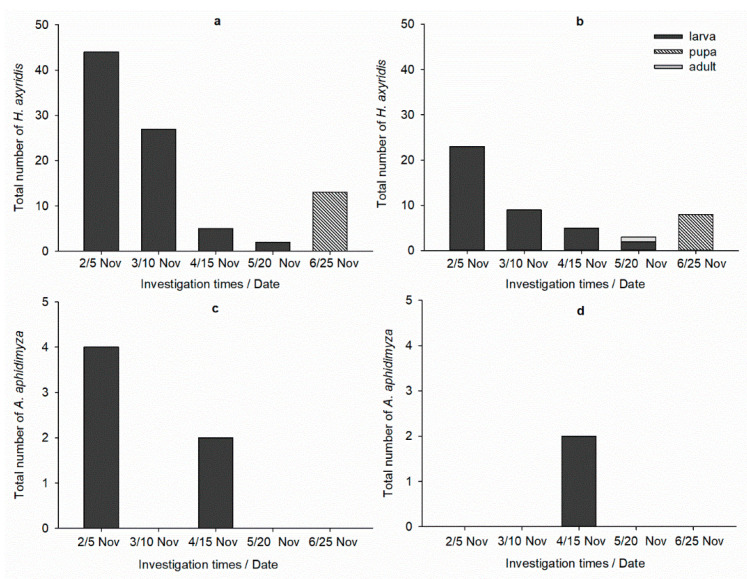
Number of predators in each treatment under low temperature. (**a**) *H. axyridis*, (**b**) low-dose thiacloprid and *H. axyridis*, (**c**) *A. aphidimyza*, (**d**) low-dose thiacloprid and *A. aphidimyza*.

**Figure 3 insects-12-00791-f003:**
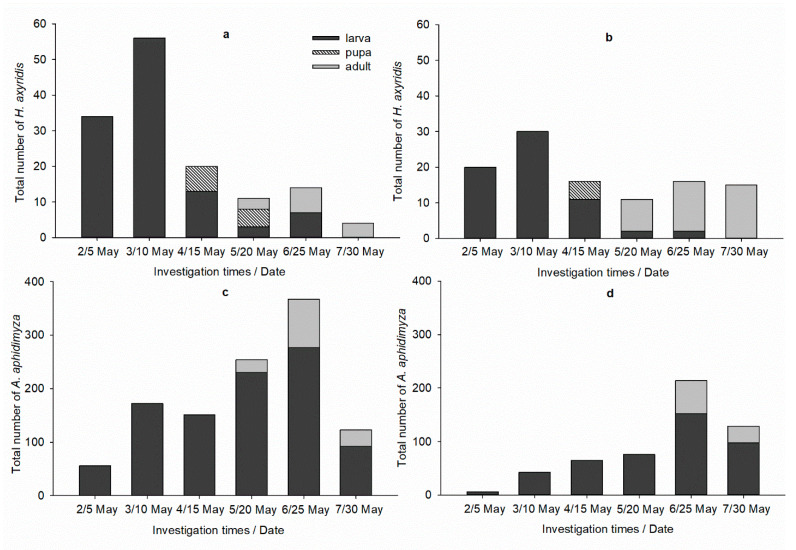
Number of predators in each treatment under high temperature. (**a**) *H. axyridis*, (**b**) low-dose thiacloprid and *H. axyridis*, (**c**) *A. aphidimyza*, (**d**) low-dose thiacloprid and *A. aphidimyza*.

**Table 1 insects-12-00791-t001:** Acute toxicity data of eight neonicotinoid pesticides tested on the aphids and the natural enemies.

Pesticides	Insects	Slope ± SE	LC_50_ (mg a.i.·L^−1^)	95% Confidence Interval (mg a.i.·L^−1^)	Correlation Coefficients (r^2^)
Imidacloprid	*M. persicae*	0.68 ± 0.064	1.847	1.019–3.304	0.900
	*H. axyridis*	0.79 ± 0.139	0.255	0.119–0.533	0.954
	*A. aphidimyza*	1.01 ± 0108	0.209	0.125–0.318	0.906
Nitenpyram	*M. persicae*	0.49 ± 0.053	0.808	0.392–1.819	0.953
	*H. axyridis*	1.67 ± 0.311	17.067	10.916–23.720	0.981
	*A. aphidimyza*	0.79 ± 0.093	0.059	0.035–0.096	0.993
Acetamiprid	*M. persicae*	1.14 ± 0.136	25.867	17.193–38.068	0.994
	*H. axyridis*	1.06 ± 0.156	0.186	0.112–0.335	0.968
	*A. aphidimyza*	1.11 ± 0.124	0.049	0.028–0.075	0.932
Thiacloprid	*M. persicae*	0.43 ± 0.048	0.043	0.019–0.096	0.911
	*H. axyridis*	1.43 ± 0.222	1.314	0.935–2.095	0.942
	*A. aphidimyza*	0.67 ± 0.072	0.128	0.067–0.232	0.995
Thiamethoxam	*M. persicae*	0.45 ± 0.054	1.927	0.859–4.792	0.995
	*H. axyridis*	0.80 ± 0.126	0.916	0.461–1.788	0.980
	*A. aphidimyza*	1.23 ± 0.130	0.116	0.077–0.168	0.928
Clothianidin	*M. persicae*	0.46 ± 0.050	0.860	0.404–1.887	0.975
	*H. axyridis*	1.04 ± 0.210	0.407	0.197–1.54	0.905
	*A. aphidimyza*	0.95 ± 0.109	0.061	0.035–0.096	0.986
Dinotefuran	*M. persicae*	0.74 ± 0.086	20.015	11.762–38.174	0.984
	*H. axyridis*	0.78 ± 0.154	0.864	0.383–3.513	0.993
	*A. aphidimyza*	0.93 ± 0.108	0.065	0.037–0.102	0.982
Flupyradifurone	*M. persicae*	0.43 ± 0.057	7.867	2.984–30.072	0.984
	*H. axyridis*	1.16 ± 0.260	2.489	1.326–3.011	0.979
	*A. aphidimyza*	0.73 ± 0.078	0.340	0.206–0.610	0.992

SE, standard error. *χ*^2^, Chi-square testing linearity of concentration-mortality responses.

**Table 2 insects-12-00791-t002:** Acute contact toxicity data of nitenpyram and thiacloprid tested on the *B. terrestris*.

Pesticides	Hours	Slope ± SE	LD_50_ (µg a.i.·bee^−1^)	95% Confidence Interval (µg a.i.·bee^−1^)	Correlation Coefficients (r^2^)
Nitenpyram	24	0.64 ± 0.073	0.592	0.314–1.229	0.975
	48	0.79 ± 0.151	0.565	0.218–1.435	
Thiacloprid	24	1.27 ± 0.251	19.825	9.254–30.352	0.981
	48	0.85 ± 0.105	17.351	6.626–28.624	

SE, standard error. *χ*^2^, Chi-square testing linearity of concentration-mortality responses.

**Table 3 insects-12-00791-t003:** Risk assessment of eight neonicotinoid pesticides to the natural enemies based on acute toxicity data and field exposure levels.

Pesticides	DT_50_ (Days)	Number of Applications	Application Interval (Days)	Recommended Application Rates (g a.i.·ha^−1^)	MAF	PER In-Field (g a.i.·ha^−1^)	LR_50_ (g a.i.·ha^−1^)	HQ In-Field	Risk	Insects
Imidacloprid	10	2	7	63.06	1.62	101.88	2.80	36.40	high	*H. axyridis*
							2.29	44.49	high	*A. aphidimyza*
Nitenpyram	10	3	10	29.99	1.75	52.48	187.45	0.28	low	*H. axyridis*
							0.65	80.99	high	*A. aphidimyza*
Acetamiprid	10	1	365	29.99	1.00	29.99	2.04	14.67	high	*H. axyridis*
							0.53	56.18	high	*A. aphidimyza*
Thiacloprid	10	2	7	9.00	1.62	14.54	14.43	1.01	low	*H. axyridis*
							1.04	10.37	high	*A. aphidimyza*
Thiamethoxam	10	2	7	56.31	1.62	90.97	10.06	9.04	high	*H. axyridis*
							1.28	71.22	high	*A. aphidimyza*
Clothianidin	10	1	365	48.00	1.00	48.00	4.47	10.74	high	*H. axyridis*
							0.67	71.18	high	*A. aphidimyza*
Dinotefuran	10	2	7	120.12	1.62	194.06	9.49	20.45	high	*H. axyridis*
							0.71	272.68	high	*A. aphidimyza*
Flupyradifurone	10	2	7	102.00	1.62	164.79	27.34	6.03	high	*H. axyridis*
							3.74	44.09	high	*A. aphidimyza*

DT_50_ is “half-life of degradation of the pesticide”, MAF is “multiple application factor”, PER in-field is “in-field predicted exposure rate”, LR_50_ is “application rate for 50% mortality”, HQ is “hazard quotient”.

**Table 4 insects-12-00791-t004:** Risk assessment of nitenpyram and thiacloprid to *B. terrestris* based on acute toxicity data.

Pesticides	Hours	Recommended Application Rates (g a.i.·ha^−1^)	LD_50_ (g a.i.·bee^−1^)	HQ In-Field	Risk
Nitenpyram	24	29.99	0.592	50.66	medium risk
	48		0.565	53.08	medium risk
Thiacloprid	24	9	19.825	0.45	low risk
	48		17.351	0.52	low risk

**Table 5 insects-12-00791-t005:** Pest control effect of *M. persicae* populations under different treatments, compared to the untreated control, on pepper seedlings set up in field cages of 90 cm × 90 cm × 90 cm within a greenhouse. L = low temperature (the dates of each investigation time were 5 November, 10 November, 15 November, 20 November, 25 November); H = high temperature (the dates of each investigation time were 5 May, 10 May, 15 May, 20 May, 25 May).

Control Measures		Investigation Times									
		2 (5th)		3 (10th)		4 (15th)		5 (20th)		6 (25th)	
		Reduction Rate ± SE	Control Effect	Reduction Rate ± SE	Control Effect	Reduction Rate ± SE	Control Effect	Reduction Rate ± SE	Control Effect	Reduction Rate ± SE	Control Effect
Control	L(Nov.)	17.44 ± 0.18	-	−96.30 ± 26.45	-	−191.42 ± 88.78	-	−148.79 ± 57.77	-	−169.00 ± 68.42	-
	H(May)	−130.85 ± 9.31	-	−204.41 ± 6.95	-	−291.00 ± 18.06	-	−461.39 ± 6.88	-	−595.28 ± 28.24	-
Thiacloprid	L(Nov.)	83.98 ± 3.22	80.63	94.85 ± 1.30	97.38	85.38 ± 1.72	94.98	78.51 ± 1.44	91.36	50.20 ± 12.08	81.45
	H(May)	94.84 ± 0.57	97.76	98.16 ± 0.22	99.39	95.95 ± 0.67	98.96	82.56 ± 2.34	96.89	2.91 ± 10.65	86.04
*H. axyridis*	L(Nov.)	63.00 ± 4.23	55.24	96.13 ± 2.30	98.03	98.58 ± 0.30	99.51	98.69 ± 0.75	99.47	99.55 ± 0.12	99.83
	H(May)	12.08 ± 2.87	61.89	90.13 ± 1.66	96.76	92.40 ± 1.84	98.06	95.65 ± 2.14	99.22	95.03 ± 2.63	99.29
*H. axyridis* + Thiacloprid	L(Nov.)	47.91 ± 1.80	36.91	78.39 ± 10.51	88.99	94.58 ± 0.76	98.13	96.11 ± 0.45	98.44	96.87 ± 0.15	98.84
	H(May)	−11.49 ± 5.05	51.68	71.94 ± 0.73	90.78	68.94 ± 1.54	92.06	51.85 ± 1.85	91.42	45.01 ± 3.10	92.10
*A. aphidimyza*	L(Nov.)	37.48 ± 0.42	24.28	4.24 ± 21.91	51.22	−45.10 ± 99.25	50.21	−11.22 ± 92.09	55.30	44.62 ± 51.81	79.41
	H(May)	−59.30 ± 6.26	30.95	16.41 ± 7.80	72.55	37.12 ± 2.49	83.94	50.12 ± 4.74	91.11	93.03 ± 1.31	99.00
*A. aphidimyza* + Thiacloprid	L(Nov.)	32.86 ± 0.36	18.68	−15.43 ± 14.86	41.20	−157.07 ± 1.49	11.79	−86.57 ± 30.09	25.01	−32.17 ± 53.64	50.87
	H(May)	−49.50 ± 10.25	35.20	−11.15 ± 6.35	63.50	−10.73 ± 6.65	71.70	6.11 ± 6.74	83.27	84.64 ± 2.64	97.79

**Table 6 insects-12-00791-t006:** Two-way ANOVA of treatments and temperatures on the population number of *M. persicae*.

Investigation Times	Factor	*df*	F	*p*
1	Treatments	5	0.13	0.98
	Temp	1	74.43	<0.0001
	Treatments * Temp	5	0.61	0.69
2	Treatments	5	63.42	<0.0001
	Temp	1	187.99	<0.0001
	Treatments * Temp	5	15.35	<0.0001
3	Treatments	5	247.21	<0.0001
	Temp	1	6.30	0.012
	Treatments * Temp	5	9.45	<0.0001
4	Treatments	5	238.10	<0.0001
	Temp	1	15.97	<0.0001
	Treatments * Temp	5	22.81	<0.0001
5	Treatments	5	540.47	<0.0001
	Temp	1	19.46	<0.0001
	Treatments * Temp	5	105.14	<0.0001
6	Treatments	5	613.78	<0.0001
	Temp	1	71.72	<0.0001
	Treatments * Temp	5	143.99	<0.0001

## Data Availability

All data generated or analyzed during this study are included in this published article.

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
