# Peer review of "Myzus persicae Management through Combined Use of Beneficial Insects and Thiacloprid in Pepper Seedlings"

_insects, 2021, doi:10.3390/insects12090791_

Round 1
Reviewer 1 Report
In this paper, the Authors present an analysis on the combined effect of chemical and biological control of Myzus persicae. In my opinion, the research has been conducted fairly well, but the presentation of results is quite rough. English language must be improved, and so the presentation of data especially in tables. The statistical analysis section must also be improved. Many statements are unclear and need a better description.
Specific comments:
Line 80: remove “adults”: females are, by definition, adults.
Line 140: please provide details about the concentration gradients.
Lines 142-143: please explain why bumblebees were kept at different conditions.
Lines 168-169: what do you mean by 0.5 larva per seedilng? You can’t use decimals for number of insects…please clarify.
Line 186: Please provide details, and not just a citation. Population of what? I presume of aphids, however please clarify. More generally, all of the statistical analyses conducted need a better description. About 2-way ANOVA: were the data normally distributed and with equal variances?
Lines 193-195: It appears to me that none of the regression equations are significant: please state it.
Table 2: It seems to me that these statistics are not about a comparison between the 2 active ingredients: they describe the mortality of bumblebees after 24 h and 48 h when exposed at different concentrations of active ingredients, according to the (quite unclear) method section. The first active ingredienti is missing in the Table body.
In general: please improve readability of table(s). A reviewer is distracted by this unclear layout and cannot correctly focus on results. You may increase column width, or use proper abbreviations.
Lines 269-270: The effectiveness of neonics on M. persicae has been demonstrated well before this paper draft, I suppose...
Lines 316-324: please italicize all of the scientific names.
Author Response
Thank you for your comments about our manuscript, we have studied comments carefully and made correction or explanation which we hope can improve the expression and accuracy of this manuscript. The main corrections and responds to comments are as flowing:
In this paper, the Authors present an analysis on the combined effect of chemical and biological control of Myzus persicae. In my opinion, the research has been conducted fairly well, but the presentation of results is quite rough. English language must be improved, and so the presentation of data especially in tables. The statistical analysis section must also be improved. Many statements are unclear and need a better description.
Answer: Thanks for your suggestion, we have invited a native English speaker have been to improve and polished this revision.
Specific comments:
Line 80: remove “adults”: females are, by definition, adults.
Answer: Thank you. Done.
Line 140: please provide details about the concentration gradients.
Answer: Thank you. We provided concentration gradients in supplement tableS1.
Lines 142-143: please explain why bumblebees were kept at different conditions.
Answer: B. terrestris usually live in the dark, and they were kept in dark condition to prevent hyperactivity.
Lines 168-169: what do you mean by 0.5 larva per seedilng? You can’t use decimals for number of insects…please clarify.
Answer: There were 30 pepper seedlings and 15 newly hatched larva of H. axyridis in one cage. We changed.
Line 186: Please provide details, and not just a citation. Population of what? I presume of aphids, however please clarify. More generally, all of the statistical analyses conducted need a better description. About 2-way ANOVA: were the data normally distributed and with equal variances?
Answer: Thank you. The calculated formulae and ‘M. persicae population’ were added. The statistical analyses details were also added, the data were normally distributed and with equal variances.
Lines 193-195: It appears to me that none of the regression equations are significant: please state it.
Answer: We used correlation coefficients (r2), and all r2 were over 90%, thus the regression equations fit the data and linear regression models.
Table 2: It seems to me that these statistics are not about a comparison between the 2 active ingredients: they describe the mortality of bumblebees after 24 h and 48 h when exposed at different concentrations of active ingredients, according to the (quite unclear) method section. The first active ingredienti is missing in the Table body.
Answer: Table 2 just show the mortality of bumblebees without comparison, and the first pesticide ‘Nitenpyram’ was added.
In general: please improve readability of table(s). A reviewer is distracted by this unclear layout and cannot correctly focus on results. You may increase column width, or use proper abbreviations.
Answer: Thank you. The tables were improved by increase column width.
Lines 269-270: The effectiveness of neonics on M. persicae has been demonstrated well before this paper draft, I suppose...
Answer: Thank you. We made revision: ‘Although, overreliance on neonicotinoids would led to the development of resistance in M. persicae population, the neonics are effective to control M. persicae in this study’.
Lines 316-324: please italicize all of the scientific names.
Answer: Thank you. Done.
Reviewer 2 Report
The subject is an area of great interest and speculation in pest management. Speculation because it is not considered a sound way to make PM recommendations. I therefore commend the authors for taking a leap of faith in trying to bring this issue (joint use of pesticides and natural enemies) as a toolbox component in pest management of aphids.
The paper needs serious grammatical review. In its present form it cannot be accepted for publication. The poor editorial work makes some of the essence of the results incomprehensible. There are facilities for editing available online, if the journal does not provide this service. I fully understand the language barrier the authors must scale to publish in an international and highly respected journal, but that cannot be compromised.
The problem starts with the abstract where broad generalizations are made, for example in the last sentence of the abstract.
Lines 32-34: If H. axyridis can maintain 90% control, why is it necessary to combine it with an insecticide? This would be considered absolutely unnecessary but most experts and thus makes the study only of academic interest. This reviewer subscribes to this school of thought. We do not need more insecticides in the environment!
Line 56: Neonicotinoids are already referred to as "neonics" in the literature; we do not need another abbreviation. Just use what is already in use.
Lines 195-197: This statement is not supported by the data presented. Authors should revisit the data in Table 1 and make the right statement in that regard.
Lines 232-239: It is well-known that rate of activity increases with temperature so this is not new. Insecticides, like most resistance phenomena, increase with temperature as the insects become more active. Lower temperature do the reverse. So why is this news?
Trying to explain these results that appear to be on a continuum is difficult and somewhat inaccurate. I think a range of temperature profile studies should be conducted for each of the components separately and then try to combine them after establishing the optimal range for each. The lack of this understanding is a weakness that flaws the experimental design. Maybe the authors did this or the information is available, in which case they need to write the methods better and discuss the results in light of this information.
Tables should be better organized. The headings should be made smaller to fit in and avoid over running the page.
RH, timing of pesticide application and diurnal behavior of the aphids and parasitoids were not mentioned in the paper. Al of these are important elements that can undermine the kind of broad recommendations the authors have concluded with. It is risky business!
Did the authors consider what a low concentration of the insecticide may do to provide a corridor for the development of resistance? I suggest they reconsider their recommendations.
What is the established action threshold (direct and indirect damage) of this pest? It seems as this would be a good place to justify why the study is necessary when H. axyridis take care of 90% of the pest. Think about that!
Author Response
Thank you for your comments about our manuscript, we have studied comments carefully and made correction or explanation which we hope can improve the expression and accuracy of this manuscript. The main corrections and responds to comments are as flowing:
Lines 32-34: If H. axyridis can maintain 90% control, why is it necessary to combine it with an insecticide? This would be considered absolutely unnecessary but most experts and thus makes the study only of academic interest. This reviewer subscribes to this school of thought. We do not need more insecticides in the environment!
Answer: We agree with your opinion, however the price of H. axyridis is expensive. Thus the combination use of predators and insecticides would be economical and effective.
Line 56: Neonicotinoids are already referred to as "neonics" in the literature; we do not need another abbreviation. Just use what is already in use.
Answer: Thank you. Done.
Lines 195-197: This statement is not supported by the data presented. Authors should revisit the data in Table 1 and make the right statement in that regard.
Answer: Thank you. The tables were improved by increase column width.
Lines 232-239: It is well-known that rate of activity increases with temperature so this is not new. Insecticides, like most resistance phenomena, increase with temperature as the insects become more active. Lower temperature do the reverse. So why is this news?
Answer: Thanks for your attention, we have corrected the related content.
Trying to explain these results that appear to be on a continuum is difficult and somewhat inaccurate. I think a range of temperature profile studies should be conducted for each of the components separately and then try to combine them after establishing the optimal range for each. The lack of this understanding is a weakness that flaws the experimental design. Maybe the authors did this or the information is available, in which case they need to write the methods better and discuss the results in light of this information.
Answer: We did not consider the effect of ‘temperature’ when we design the experiment at first. We just conducted the greenhouse trails at higher temperature when we found the control effect of A. aphidimyza was not as expected due to the low temperature.
Tables should be better organized. The headings should be made smaller to fit in and avoid over running the page.
Answer: Thank you. Done.
RH, timing of pesticide application and diurnal behavior of the aphids and parasitoids were not mentioned in the paper. Al of these are important elements that can undermine the kind of broad recommendations the authors have concluded with. It is risky business!
Answer: In the greenhouse trails the pesticide was sprayed at 4:00 p.m. at the density of 1 aphid per plant. There were no parasitoids during the trails, but previous studies have shown that the efficacy of aphid control by A. aphidimyza would be improved when combined with parasitoids Aphidius colemani (mentioned in discussion).
Did the authors consider what a low concentration of the insecticide may do to provide a corridor for the development of resistance? I suggest they reconsider their recommendations.
Answer: We think the low concentration insecticide can reduce the activity of aphids, then the aphids could be more vulnerable searched and eaten by predators. When most of the aphids were eaten, the resistance would be negligible.
What is the established action threshold (direct and indirect damage) of this pest? It seems as this would be a good place to justify why the study is necessary when H. axyridis take care of 90% of the pest. Think about that!
Answer: The action threshold are significantly ranged due to growing season and crops. The M. persicae population grows rapidly in few days, so we think it is better to take action at once when find the aphids. In this study, the action threshold of M. persicae was 1 aphid / plant.
Reviewer 3 Report
This is an interesting paper of extensive work. However, the level of written English really lets the manuscript down. The Simple Summary alone has sentences such as ‘no matter combination with thiacloprid or not at above 20°C condition’ and, as a native English speaker, I have difficulty in understanding what the basic summary is trying to tell me. The concluding sentence of both the Simple Summary and the Abstract state ‘Our results indicated that it is necessary to choose H. axyridis or A. aphidimyza flexibly to control aphids via comprehensive consideration of environmental conditions and control costs’. This is incredibly vague and I am not entirely sure what the study is suggesting.
The Introduction is very interesting and covers the necessary background information, with up-to-date information on the current use of Neonicotinoid insecticides and interesting information on the biological control agents selected for study. However, once again, the level of English must be addressed. Furthermore, in L86-87 you leap between discussing A. aphidimyza and then bumblebee pollination. There needs to be more of a connection and explanation between these two seemingly-disconnected topics.
L101: what do you mean by mixed population? This could use some explanation. Do you mean a mixture of alate and apterous individuals? Different life stages?
L143: why was Bombus terrestris kept under continuous darkness? Perhaps this needs explanation.
L157: is this equation referred to in the text?
What were the treatment temperatures? What low temperature was used and what high temperature was used? Where these constant or fluctuating? The first reference I see of this is in L226 in the Results section.
What statistical package was used to perform the analyses?
Author Response
Thank you for your comments about our manuscript, we have studied comments carefully and made correction or explanation which we hope can improve the expression and accuracy of this manuscript. The main corrections and responds to comments are as flowing:
This is an interesting paper of extensive work. However, the level of written English really lets the manuscript down. The Simple Summary alone has sentences such as ‘no matter combination with thiacloprid or not at above 20°C condition’ and, as a native English speaker, I have difficulty in understanding what the basic summary is trying to tell me. The concluding sentence of both the Simple Summary and the Abstract state ‘Our results indicated that it is necessary to choose H. axyridis or A. aphidimyza flexibly to control aphids via comprehensive consideration of environmental conditions and control costs’. This is incredibly vague and I am not entirely sure what the study is suggesting.
Answer: Thank you. We made revision: ‘Both H. axyridis and A. aphidimyza effectively controlled aphids whether combined with thiacloprid, at above 20℃ condition’, ‘Our results indicated that it is cost-effective to control M. persicae with A. aphidimyza at suitable temperature condition, H. axyridis was more effective at low temperature’.
The Introduction is very interesting and covers the necessary background information, with up-to-date information on the current use of Neonicotinoid insecticides and interesting information on the biological control agents selected for study. However, once again, the level of English must be addressed. Furthermore, in L86-87 you leap between discussing A. aphidimyza and then bumblebee pollination. There needs to be more of a connection and explanation between these two seemingly-disconnected topics.
Answer: Thank you. We made revision: ‘Another ecological agricultural technical measure —’the bumblebee pollination technique’ can increase the quality and yield of pollinated crops which is well-suited for greenhouse crops.’
L101: what do you mean by mixed population? This could use some explanation. Do you mean a mixture of alate and apterous individuals? Different life stages?
Answer: It means different life stages. Done.
L143: why was Bombus terrestris kept under continuous darkness? Perhaps this needs explanation.
Answer: B. terrestris usually live in the dark, and they were kept in dark condition to prevent hyperactivity.
L157: is this equation referred to in the text?
Answer: The equation did not show in the text.
What were the treatment temperatures? What low temperature was used and what high temperature was used? Where these constant or fluctuating? The first reference I see of this is in L226 in the Results section.
Answer: The details were showed in methods of ‘greenhouse efficacy trials’. Low temperature trails were conducted from October 29 to December 19, and high temperature trails were conducted during April 28 to May 30. ‘The temperature fluctuation recorded during the tests shown in Figure S1’.
What statistical package was used to perform the analyses?
Answer: The ‘fitdistr’ and ‘AIC’ functions in R was used to identify error distributions. Then two-way ANOVA and Tukey’s Honestly Significant Differences (HSD) were used by SPASS 18.
Reviewer 4 Report
The manuscript is basically well-written and follows scientific logic and presentation. The statistics appear adequate. The Figures and Tables need to be improved and suggestions are made. There are many edits to help improve the readability of the text. The research will help in slow resistance to neonicotinoids and lower insecticide dose in greenhouse IPM programs.
The experimental design an statistics are adequate.
However the treatments and the number of replicates for each treatment needs to be better stated in the methods and table 1. The two temperatures should have bee done at the same time, but that is not possible when using only greenhouses and adding incubators creates other issues. On Table 1 add the China gov LC50 and the replicates for each insect by treatment.
Please explain if each treatment had 2 additions of predators or the predators were introduced only once to a treatment. " lime 176-177, Adults of A. aphidimyza were introduced on October 29th and November 5th while larvae of H. axyridis were introduced on November 1st and November 5th".
The supplemental data on temp fluctuations needs to be included in the text in one sentence as mean temp by sampling date and mean over all dates for the 2 treatments. The graph of temperature will then be unnecessary.
There are many suggestions to clarify the wording and data throughout the MS.

Author Response
Thank you for your comments about our manuscript, we have studied comments carefully and made correction or explanation which we hope can improve the expression and accuracy of this manuscript. The main corrections and responds to comments are as flowing:
However the treatments and the number of replicates for each treatment needs to be better stated in the methods and table 1. The two temperatures should have bee done at the same time, but that is not possible when using only greenhouses and adding incubators creates other issues. On Table 1 add the China gov LC50 and the replicates for each insect by treatment.
Answer: Thank you. We stated the replicates in methods: ‘each treatment with three biological replications’, and we remain unchanged in Table1 since it would be miscellaneous when stated in Table1. As for the greenhouses trails, we did not consider effects of temperature at first, so when we found the control effect of A. aphidimyza was not as expected due to the low temperature, we then conducted the trails at higher temperature.
Please explain if each treatment had 2 additions of predators or the predators were introduced only once to a treatment. " lime 176-177, Adults of A. aphidimyza were introduced on October 29th and November 5th while larvae of H. axyridis were introduced on November 1st and November 5th".
Answer: A. aphidimyza and H. axyridis were introduced to different treatments, and each treatment introduced 1 kind of predators twice. And considering A. aphidimyza developed slowly at low temperature condition, we introduced adults of A. aphidimyza earlier.
The supplemental data on temp fluctuations needs to be included in the text in one sentence as mean temp by sampling date and mean over all dates for the 2 treatments. The graph of temperature will then be unnecessary.
Answer: Thank you. We added the mean temp over all dates, and retained the Fig. S1.
There are many suggestions to clarify the wording and data throughout the MS.
Answer: Thank you. We tried our best to polish the wording and the tables.
Round 2
Reviewer 1 Report
I feel that the main concerns that I have raised in the first revision have been adequatelly addressed by the Authors. Yet, there are still a few matters to be fixed: in particular, I can't see anywhere Table S1 showing the concentration gradients as stated in Line 140 (attachment missing? By clicking on Supplementary files, I've downloaded only Figure S1). Also, I would like to see Tables 2 and 5 in their clean, final form (without track changes), so I could read them properly.
Thank you
Author Response
Thank you for your comments about our manuscript, we have studied comments carefully and made correction or explanation which we hope can improve the expression and accuracy of this manuscript. We have invited DBMediting for professional english language editing services.
Reviewer 2 Report
I do not agree with some of the responses from the authors, but I will not hold back the manuscript because of my perceptions, for example the possible out come of the use of low concentration of insecticide and likelihood of resistance.
Author Response

(The authors gave the same response as above.)

Reviewer 4 Report
The text needs considerable revision based on its use of English. It is currently very difficult to read. Numerous suggestions were made, but were not fixed. The author needs to change the text and not only answer the reviewer's questions.
Changing NI to neonics is not correct. Use the proper word neonicotinoids.
Author Response

(The authors gave the same response as above.)

Round 3
Reviewer 4 Report
The concept and the experimental design is adequate. However, the manuscript still needs more work in making it easier to read and to follow the science.
The English was improved, but further improvement in sections is still needed. Sometimes the sentences prevented the reader from understanding the science, such as lines 265-269.
Change the title from to “Aphid management through combined use of beneficial insects and thiacloprid in pepper seedlings”
Do not use neutral Insect or non-pest insect or other words; only use pest and beneficial insects
Remove sustainable as it is not defined; the current issue is aphid population management and the variable that you measure is population size, not dynamics or development.
Change gradient to dose; you use 6 doses in a gradient. Also, use only one word dose and not gradient and concentration it is confusing.
The sampling dates were added but they need proper scientific citation. Start with 24 hrs. = DAT days after treatment 1, etc. Put these DAT number first and add the J dates in parentheses.
Population dynamics/development etc. all need to be changes to the same word population size. Figure 1 population size (not development).
Instead of using the word survey use DAT days after treatment. Are these data combined for the 2 replicate experiments?
Table 1 is unreadable due to the edits. Please provide the fixed version only.
Figure 1 put the high and low temp mean in parentheses
Figures 2 and 3 AND lines 170-178 Different numbers of midges and lady beetles were added to the treatments. So the numbers should be the net gain. Otherwise they cannot be compared.
In supplemental data Table S1 and S2 add the number of insects in each rep and the numbers of reps in the table legend. Also change gradient to dose.
Suggestion Acute LC50 bioassays were performed with 5 or 6 doses or 8 neonicotinoid insecticides using x insects /cage and 3 reps
Line 55 selectivity in spatial temporal. Remove as you are not discussing this concept in the paper
Line 68 use pest inset and beneficial insect, no other general terms
Line 135 larvae (plural) were
Line 209-210 which neonicotinoid? The line does not say
Lines 213-214 for the bees add the LC50 as was done above for the midges and lady beetles
Line 243 6 surveys? use DAT
Line 265-268 not readable; please edit
Line 269 only 4 left, how many did you start with; use a percentage
Line 296-306 the English is difficult to understand
Line 318 do not capitalize Thiacloprid
Line 326 Where did n=367 come from? That information is needed earlier in the methods. You need to tell the reader the n= number of insects used in each treatment.
Line 335 -336 these data contradicts the data in the previous sentence. Also, there is no mention of egg hatching? Where is that data? Use number of larvae as in Figures 2 and 3.
I did not edit the MS for English as last time I spent alot of time doing that and none of the edits were used. So there are no comments on the manuscript.
Author Response
The concept and the experimental design is adequate. However, the manuscript still needs more work in making it easier to read and to follow the science.
The English was improved, but further improvement in sections is still needed. Sometimes the sentences prevented the reader from understanding the science, such as lines 265-269.
Change the title from to “Aphid management through combined use of beneficial insects and thiacloprid in pepper seedlings”
Answer: Thank you. We changed.
Do not use neutral Insect or non-pest insect or other words; only use pest and beneficial insects
Answer: Thank you. Done.
Remove sustainable as it is not defined; the current issue is aphid population management and the variable that you measure is population size, not dynamics or development.
Answer: We replaced ‘sustainable’ with ‘lasting’.
Change gradient to dose; you use 6 doses in a gradient. Also, use only one word dose and not gradient and concentration it is confusing.
Answer: We deleted ‘gradient’, just keep ‘concentrations’. And we guess ‘dose’ refers to ‘low-dose’, all ‘low dose’ in MS were concentration dose of LC50, we added ‘the concentration of LC50’ in methods of ‘Greenhouse efficacy trial’.
The sampling dates were added but they need proper scientific citation. Start with 24 hrs. = DAT days after treatment 1, etc. Put these DAT number first and add the J dates in parentheses.
Answer: The first application time were different among the treatments, so it would be confused to use DAT days. We think the dates would be more clear.
Population dynamics/development etc. all need to be changes to the same word population size. Figure 1 population size (not development).
Answer: Thank you. Done.
Instead of using the word survey use DAT days after treatment. Are these data combined for the 2 replicate experiments?
Answer: The ‘Greenhouse efficacy trial’ were conducted two times, each had 2 replicates, and the replicates date were combined. So it would be more difficult to understand if changed ‘survey’ with date, we added dates in parentheses.
Table 1 is unreadable due to the edits. Please provide the fixed version only.
Answer: Thank you. Done.
Figure 1 put the high and low temp mean in parentheses
Answer: Thank you. Done.
Figures 2 and 3 AND lines 170-178 Different numbers of midges and lady beetles were added to the treatments. So the numbers should be the net gain. Otherwise they cannot be compared.
Answer: There would be negative value if use the net gain, and we just compare the numbers of same treatment at different temperature, not different treatments.
In supplemental data Table S1 and S2 add the number of insects in each rep and the numbers of reps in the table legend. Also change gradient to dose.
Answer: Thank you. Done.
Suggestion Acute LC50 bioassays were performed with 5 or 6 doses or 8 neonicotinoid insecticides using x insects /cage and 3 reps
Answer: Thank you. Done.
Line 55 selectivity in spatial temporal. Remove as you are not discussing this concept in the paper
Answer: Thank you. Done.
Line 68 use pest inset and beneficial insect, no other general terms
Answer: Thank you. Done.
Line 135 larvae (plural) were
Answer: Thank you. Done.
Line 209-210 which neonicotinoid? The line does not say
Answer: A. aphidimyza was sensitive to the eight neonicotinoids, all LC50 ≤ 0.34 mg a.i.·L−1.
Lines 213-214 for the bees add the LC50 as was done above for the midges and lady beetles
Answer: Thank you. We added as ‘the LD50 were < 0.6 µg a.i.·bee-1 and > 17 µg a.i.·bee-1 respectively’.
Line 243 6 surveys? use DAT
Answer: Thank you. We used ‘investigation’ and added dates in parentheses.
Line 269 only 4 left, how many did you start with; use a percentage
Answer: The number of H. axyridis was 30 per cage at first, and there were only 4 adults in two cages at last. We add ‘there were only 4 out of 60’.
Line 296-306 the English is difficult to understand
Answer: Thank you. We modified.
Line 318 do not capitalize Thiacloprid
Answer: Thank you. Done.
Line 326 Where did n=367 come from? That information is needed earlier in the methods. You need to tell the reader the n= number of insects used in each treatment.
Answer: Thank you. We added the information in methods, ‘The total number of larvae, pupae and adults of H. axyridis and A. aphidimyza in each treatment were counted.’
Line 335 -336 these data contradicts the data in the previous sentence. Also, there is no mention of egg hatching? Where is that data? Use number of larvae as in Figures 2 and 3.
Answer: We discussed the effect of temperature on A. aphidimyza by citing previous studies which are consistent with our findings, so we did not add our own data.
I did not edit the MS for English as last time I spent a lot of time doing that and none of the edits were used. So there are no comments on the manuscript.
Answer: Sorry, we did not receive edited MS last time.